# Expanding the Molecular Spectrum of *ANKRD11* Gene Defects in 33 Patients with a Clinical Presentation of KBG Syndrome

**DOI:** 10.3390/ijms23115912

**Published:** 2022-05-25

**Authors:** Ilaria Bestetti, Milena Crippa, Alessandra Sironi, Francesca Tumiatti, Maura Masciadri, Marie Falkenberg Smeland, Swati Naik, Oliver Murch, Maria Teresa Bonati, Alice Spano, Elisa Cattaneo, Milena Mariani, Fabio Gotta, Francesca Crosti, Pietro Cavalli, Chiara Pantaleoni, Federica Natacci, Maria Francesca Bedeschi, Donatella Milani, Silvia Maitz, Angelo Selicorni, Luigina Spaccini, Angela Peron, Silvia Russo, Lidia Larizza, Karen Low, Palma Finelli

**Affiliations:** 1Laboratory of Medical Cytogenetics and Molecular Genetics, IRCCS Istituto Auxologico Italiano, 20142 Milan, Italy; m.crippa@auxologico.it (M.C.); a.sironi@auxologico.it (A.S.); f.tumiatti@auxologico.it (F.T.); maura.masciadri@alice.it (M.M.); s.russo@auxologico.it (S.R.); l.larizza@auxologico.it (L.L.); 2Department of Medical Biotechnology and Translational Medicine, University of Milan, 20142 Milan, Italy; 3Department of Medical Genetics, University Hospital of North Norway, 9019 Tromsø, Norway; marie.smeland@unn.no; 4Clinical Genetics Unit, Birmingham Women’s Hospital, Birmingham B15 2TG, UK; swati.naik@nhs.net; 5All Wales Medical Genomics Service, University Hospital of Wales, Cardiff CF14 4XW, UK; oliver.murch@wales.nhs.uk; 6Clinic of Medical Genetics, San Luca Hospital, IRCCS Istituto Auxologico Italiano, 20142 Milan, Italy; mariateresa.bonati@burlo.trieste.it; 7Clinical Pediatric Genetic Unit, Pediatric Clinic, Fondazione MBBM, San Gerardo Hospital, 20900 Monza, Italy; alice.spano@maggioreosp.novara.it (A.S.); maitz.silvia@gmail.com (S.M.); 8Clinical Genetics Unit, Department of Obstetrics and Gynecology, “V. Buzzi” Children’s Hospital, University of Milan, 20142 Milan, Italy; elisa.cattaneo@asst-fbf-sacco.it (E.C.); luigina.spaccini@asst-fbf-sacco.it (L.S.); 9Pediatric Unit, ASST Lariana, 22100 Como, Italy; milena.mariani@asst-lariana.it (M.M.); angelo.selicorni61@gmail.com (A.S.); 10Clinical Genetics, ASST Cremona, Via Concordia 1, 26100 Cremona, Italy; gotta.fabio@libero.it (F.G.); pietro.cavalli@gmail.com (P.C.); 11Medical Genetics Laboratory, Clinical Pathology Department, S. Gerardo Hospital, 20900 Monza, Italy; f.crosti@asst-monza.it; 12Department of Pediatric Neuroscience, Fondazione IRCCS Istituto Neurologico Carlo Besta, 20142 Milan, Italy; chiara.pantaleoni@istituto-besta.it; 13Medical Genetic Unit, Fondazione IRCCS Ca’ Granda Ospedale Maggiore Policlinico, 20142 Milan, Italy; federica.natacci@policlinico.mi.it (F.N.); mariafrancesca.bedeschi@policlinico.mi.it (M.F.B.); 14Pediatric Highly Intensive Care, Fondazione IRCCS Ca’ Granda Ospedale Maggiore Policlinico, 20142 Milan, Italy; donatella.milani@policlinico.mi.it; 15Service of Medical Genetics, Oncologic Institute of Southern Switzerland, EOC, 6900 Lugano, Switzerland; 16Child Neuropsychiatry Unit-Epilepsy Center, Department of Health Sciences, ASST Santi Paolo e Carlo, San Paolo Hospital, Università Degli Studi di Milano, 20142 Milan, Italy; angela.peron@unimi.it; 17Medical Genetics, ASST Santi Paolo e Carlo, San Paolo Hospital, 20142 Milan, Italy; 18Division of Medical Genetics, Department of Pediatrics, University of Utah School of Medicine, Salt Lake City, UT 84132, USA; 19University Hospitals Bristol NHS Trust, University of Bristol, Bristol BS1 3NU, UK; karen.low@uhbw.nhs.uk

**Keywords:** KBG syndrome, *ANKRD11* variations, diagnostic flow chart, *ANKRD11* gene expression analysis

## Abstract

KBG syndrome (KBGS) is a neurodevelopmental disorder caused by the Ankyrin Repeat Domain 11 (*ANKRD11*) haploinsufficiency. Here, we report the molecular investigations performed on a cohort of 33 individuals with KBGS clinical suspicion. By using a multi-testing genomic approach, including gene sequencing, Chromosome Microarray Analysis (CMA), and RT-qPCR gene expression assay, we searched for pathogenic alterations in *ANKRD11*. A molecular diagnosis was obtained in 22 out of 33 patients (67%). *ANKRD11* sequencing disclosed pathogenic or likely pathogenic variants in 18 out of 33 patients. CMA identified one full and one terminal *ANKRD11* pathogenic deletions, and one partial duplication and one intronic microdeletion, with both possibly being pathogenic. The pathogenic effect was established by RT-qPCR, which confirmed *ANKRD11* haploinsufficiency only for the three deletions. Moreover, RT-qPCR applied to six molecularly unsolved KBGS patients identified gene downregulation in a clinically typical patient with previous negative tests, and further molecular investigations revealed a cryptic deletion involving the gene promoter. In conclusion, *ANKRD11* pathogenic variants could also involve the regulatory regions of the gene. Moreover, the application of a multi-test approach along with the innovative use of RT-qPCR improved the diagnostic yield in KBGS suspected patients.

## 1. Introduction

KBG syndrome (KBGS; OMIM #148050) is an autosomal dominant neurodevelopmental disorder (NDD) first described in 1975 [1] that is caused by the haploinsufficiency of the *ANKRD11* gene (OMIM #611192) at 16q24.3 due to heterozygous pathogenic variants or chromosomal imbalances [2,3,4]. KBGS is characterized by a distinctive facial appearance, macrodontia of upper central permanent incisors, short stature, hearing loss, developmental delay, learning difficulties, and neurobehavioral problems. Other features that can further help to establish a clinical diagnosis are seizures, brachydactyly or relevant hand anomalies, cryptorchidism in males, feeding problems, palatal abnormalities, and delayed anterior fontanelle closure [5].

Due to significant variability in the phenotype, even among affected individuals of the same family, a clinical diagnosis of KBGS may be easily missed. Indeed, some patients may show very subtle or unrecognizable clinical features, as demonstrated by the recent large-scale Developmental Delay Disorders (DDD) studies that revealed *ANKRD11* as one of the most frequently mutated gene in patients with a severe neurodevelopmental disorder that was initially not attributed to KBGS [6,7,8,9]. In addition, a patient carrying *ANKRD11* deletion, with normal psychomotor development and clinical signs reminiscent of the Silver–Russell syndrome ((SRS; OMIM#180860), has been reported [10]. The diagnosis is even more arduous since several neurodevelopmental disorders such as the Cornelia de Lange (CdLS; OMIM#122470), Coffin–Siris (CSS; OMIM#135900), and Mental Retardation Autosomal Dominant 23 (MRD23; OMIM#615761) syndromes share some phenotypic features with KBGS, including a variable degree of intellectual disability and developmental delay, growth retardation, and common facial dysmorphisms [11,12,13,14]. This phenotypic overlap appears to reflect the molecular interaction of the causative genes of these syndromes, all acting as chromatin modifiers in common cellular mechanisms and pathways [15].

*ANKRD11* encodes for a crucial chromatin co-regulator that controls histone acetylation and gene expression during neural development by recruiting chromatin remodelers upon interaction with specific transcriptional repressors or activators [16,17]. This is due to its unique protein structure, which contains two repression domains and one activation domain [18,19]. To date, more than 200 individuals harboring heterogeneous *ANKRD11* molecular defects have been reported [20]. Truncated pathogenic variants (frameshift or nonsense) are prevalent and cluster mostly in exon 9, possibly because it represents more than 80% of the coding region, while disease-causing missense variants are uncommon. Pathogenic deletions involving the entire gene or parts of *ANKRD11* have been frequently described [21,22]. A single intragenic *ANKRD11* duplication segregating in a family was reported and characterized as pathogenic [3].

Given the molecular spectrum and broad phenotypic variability of KBGS, the optimal diagnostic approach is via multi-testing based on *ANKRD11* sequencing—by single gene test or multi-gene panels—and chromosome microarray analysis (CMA) [23].

Here, we report the clinical and molecular findings in 33 patients with a KBGS/KBGS-like clinical diagnosis and confirm the utility of a “multi-testing” approach for molecular diagnosis. In addition to the standard tools, we applied an RT-qPCR *ANKRD11* expression analysis as a new strategy to detect *ANKRD11* haploinsufficiency caused by cryptic deletions and to highlight the clinical relevance of sub-microscopic rearrangements.

## 2. Results

In order to disclose the molecular alterations underlying the patients’ phenotype, we developed a flowchart, illustrated in Figure 1, based on combined genomic approaches.

### 2.1. Sequence Analysis Discovered New ANKRD11 Pathogenic Variants

*ANKRD11* sequencing was performed on all but two patients, revealing 23 distinct *ANKRD11* variants in 23 out of 31 patients (Table 1). All variants were mapped throughout exon 9, except for one located in exon 11 (PT27). Of the sequenced cases, the molecular diagnosis was confirmed in 18 patients, whose clinical characteristics are reported in Appendix A. The origin of the variants could be assessed to be de novo in 11 out of 18 patients (Table 1). In detail, nine out-of-frame deletions, seven nonsense variants, and one out-of-frame duplication leading to premature stop codons in the *ANKRD11* exon 9 sequence were found and classified as pathogenic according to the ACMG guidelines (Table 1 and Appendix A). These pathogenic variants are predicted to cause either truncated proteins with different lengths and domain compositions or the activation of nonsense-mediated decay leading to gene haploinsufficiency. A missense variant involving exon 11 was identified in PT27 and classified as likely pathogenic. A possible splicing alteration has been predicted for this variant by in silico analysis performed with the Human Splicing Finder tool [24]. Based on the prediction, the SNV might activate an exonic cryptic donor site, causing the skipping of 36 residues encoded by exon 11 and the introduction of 30 aa downstream of a premature stop codon in exon 12 (Appendix A). Moreover, this variant maps to the C-terminal region of the protein, involving the residues from 2369 to 2663, which are important for its degradation. This region is fully absent in the predicted truncated protein.

In the remaining five patients, six missense variants inherited from an unaffected parent were identified, and none of them was classified as pathogenic (Table 1 and Appendix A). Based on the ACMG guidelines, four were considered to be likely benign, while the two variants found in the same patient were classified as of uncertain significance and benign, respectively (Table 1 and Appendix A). No splice site alterations have been predicted for these variants.

Overall, 12 out of the 23 variants were previously unreported in KBGS patients, while three deletions, namely p.(Lys635GlnfsTer26), p.(Lys803ArgfsTer5), and p.(Glu461GlnfsTer48), found in patients 17, 18, 22, and 23, are recurrent pathogenic variants (Table 1) [5,19,21]. The variants identified in the present study are publicly available at the ClinVar database (http://www.ncbi.nlm.nih.gov/clinvar/ (accessed on 15 February 2022)): accession numbers SCV002097343–SCV002097365.

### 2.2. High-Resolution CMA Evidenced Never Reported ANKRD11 Rearrangements

High-resolution CMA was performed on 12 patients, 10 of whom were negative to sequence analysis, while 2 reported this to be their first molecular test. Four rare CNVs involving *ANKRD11* were identified in PT4, PT12, PT13, and PT33 and classified as pathogenic (PT4 and PT33) or likely pathogenic (PT12 and PT13) (Table 2). The clinical characteristics of the four patients are reported in Appendix A, and facial appearances of PT4 and PT33 are shown in Figure 2.

In detail, a de novo pathogenic deletion of 27.5 kb was detected within 16q24.3 (chr16:89,307,972-89,335,487, hg19) in PT4, including the terminal part of the *ANKRD11* gene from IVS 11/12, depending on the maximum/minimum deletion size (reference transcript NM_013275.6) (Table 2) (Figure 3A). Further 1M CMA analysis redefined the deletion size to 28.4 kb, narrowing the proximal and distal deletion breakpoints within regions of 3.5 kb (chr16:89,304,429-89,307,972, hg19) and 3.7 kb (chr16:89,336,366-89,340,144, hg19), respectively, with the latter encompassing *ANKRD11* exon 12.

PT12 was found to be carrying a de novo likely pathogenic microdeletion of 682 bp (chr16:89,555,339-89,556,020, hg19) within *ANKRD11* IVS1 (Table 2). The deletion partially involved the CpG island 205, which matches the gene promoter (Figure 3A).

In PT13, CMA revealed a likely pathogenic microduplication of 200 kb at 16q24.3 (chr16:89,220,725-89,420,725, hg19), partially involving *ANKRD11* from 3′UTR to IVS 2 and the genes *ACSF3*, *CDH15*, *SLC22A31*, and *ZNF778* (Table 2) (Figure 3A). A de novo origin of the duplication could not be established as only the patient’s father was available for analysis.

Finally, PT33 was found to have a pathogenic 1.2 Mb deletion at 16q24.3 (chr16:88,365,786-89,584,412, hg19), encompassing *ANKRD11* and other 11 OMIM genes.

All CNVs were submitted to ClinVar (http://www.ncbi.nlm.nih.gov/clinvar/ (accessed on 15 February 2022)): accession numbers SCV002097366–SCV002097371.

### 2.3. RT-qPCR Analysis: A Valuable Diagnostic Approach

*ANKRD11* RT-qPCR analysis was performed on 10 patients—three (PT4, PT12, and PT13) harboring *ANKRD11* genomic imbalances were chosen in order to assess their pathogenic effect, and seven molecularly unsolved patients (PT7, PT11, PT14, PT16, PT21, PT31, and PT32) were selected to investigate *ANKRD11* haploinsufficiency.

The analysis of PT4, who was harboring a terminal *ANKRD11* deletion, revealed half the amount of expected wild-type *ANKRD11* transcript and a truncated one, with a missing exon junction 12–13, thus demonstrating the pathogenicity of the deletion (Figure 3B).

*ANKRD11* haploinsufficiency was identified in two patients (Figure 3B). PT12 analysis indicated the pathogenic role of the intronic deletion, which conceivably prevents the transcription of the deleted allele due to partial loss of the promoter, whereas in PT14, in whom all previous analyses were normal, the discovery of gene haploinsufficiency pinpointed a cryptic defect, which was likely affecting the *ANKRD11* promoter (Figure 3B). The clinical characteristics of PT14 are reported in Appendix A.

In the other six patients, an *ANKRD11* mRNA level comparable to controls was detected (Figure 3B). In particular, the analysis of PT13, carrier of a partial *ANKRD11* duplication, ruled out gene deregulation caused by the CNV. Accordingly, the duplication classified as likely pathogenic at first was downgraded to VUS (variant of unknown significance).

### 2.4. Molecular Characterization of PT4 and PT14 Patients’ CNVs

Consequent to the previous findings for PT4, targeted qPCR analysis enabled the confirmation of the involvement of *ANKRD11* exon 12 in the deletion (reference transcript NM_001256182.1) (Figure 4A). The deletion bkps were thus precisely mapped at the nucleotide level by means of long-range PCR, with primers flanking the breakpoint regions, and Sanger sequencing clarifying the exact deletion length of 33,179 kb (chr16:89,306,725-89,339,903, hg19) (Figure 4B). Sequence mapping located the intragenic breakpoint within IVS12 and revealed an overlapping region of 9 bp (TCGAGACCA) at the bkp junction (Figure 4B). Finally, a low-grade mosaic deletion was excluded in the parents, at least in the blood, as no amplification of the breakpoint junction was obtained from them by long-range PCR (data not shown).

In PT14, with a supposed promoter defect causing *ANKRD11* haploinsufficiency, 1M array-CGH revealed the deletion of probe A_16P03198945 (chr16:89,557,919-89,557,978, hg19), mapping to an intergenic region 951 bp from *ANKRD11* 5′UTR. Quantitative PCR on genomic DNA with four different probes specifically designed to validate and refine the array data confirmed the presence of a cryptic deletion with probes number 2 and 3 (Figure 4A). The subsequent PCR performed with probe 1 primer forward and probe 4 primer reverse, both not deleted in PT14, detected an amplicon of about 1000 bp only in the patient. Sanger sequencing enabled the confirmation of the presence of two in cis deletions: one of 16 bp (chr16:89,556,553-89,556,568, hg19) inside IVS1 of *ANKRD11*, and the second of 1781 bp (chr16:89,556,619-89,558,399, hg19) partially involving IVS1, fully comprising exon 1 and including part of the promoter (CpG island 205) (Figure 4C). Similarly to PT4, low-level mosaicism for the deletion was excluded in the parents’ blood by means of targeted bkp junction amplification (data not shown).

### 2.5. Statistical Analysis

The analysis did not show any statistical differences between the clinical features of the molecularly confirmed and not confirmed KBGS patients (Appendix A). Borderline *P* values were obtained for three clinical signs, namely low anterior/posterior hairlines (*p* = 0.05), synophrys (*p* = 0.066), hearing loss (*p* = 0.056), which were more evident in the molecularly confirmed KBGS group.

## 3. Discussion

The number of patients with genetic defects involving *ANKRD11* described in the literature is increasingly expanding [4,8,28]. In the present study on recruited patients with a KBG or KBG-like clinical phenotype, we achieved a molecularly confirmed diagnosis of KBG syndrome in 22 out of 33 patients (67%). The diagnostic yield by *ANKRD11* sequencing was 82% (18/22), and by CMA it was 14% (3/22); these are in line with the frequencies reported in the literature [8,22]. A small increase in the detection rate was yielded by the innovative use of RT-qPCR analysis, which allowed for the definitive assessment of *ANKRD11* haploinsufficiency caused by a cryptic rearrangement in one of the 22 molecularly confirmed patients. These results reflect the frequency of *ANKRD11* genetic alterations decreasing from SNVs (69%) to full or partial gene deletions (17%), while duplications supposed or proved to interrupt the gene are extraordinarily rare in the literature and public databases [13,22,29].

With the advent of the use of high-resolution CMA with specific NDD-enriched regions, several submicroscopic rearrangements in *ANKRD11* are now emerging [23], as identified in PT12, who bears a 682 bp deletion. Most of the small deletions fall in non-coding regions of the gene, namely the first introns or the promoter region (CpG 205), suggesting their proneness to genomic rearrangements. The use of these platforms could unveil the molecular defects in patients suspected to have KBG and who are negative to NGS screening, which would otherwise remain unsolved. Nevertheless, several cryptic deletions could be undetectable with the conventional diagnostic tests, as demonstrated in the present study by the complex rearrangement of PT14. For this patient, only the application of an RT-qPCR analysis confirmed the KBG clinical diagnosis, which showed a halved quantity of transcript. This result could be ascribed to two submicroscopic deletions in exon 1 and IVS1, precisely mapped upon a deep study of the *ANKRD11* promoter regions.

Apart from identifying subtle pathogenic CNVs that would otherwise be missed, RT-qPCR analysis has proven to be a useful test for exploring the pathogenicity of genomic deletions/duplications with an unclear molecular effect. For KBG PT4 bearing a terminal *ANKRD11* deletion, the expression analysis did not show a gene downregulation but highlighted an aberrant transcript lacking the last exons of the gene. The pathogenicity of this terminal deletion was then corroborated by the stability of the aberrant transcript (not undergoing nonsense-mediated decay) predicted to encode for a dysfunctional protein lacking the RD2 C-terminal domain [19]. Similarly, for KBG PT12 who had an intronic deletion of uncertain significance, the RT-qPCR showed gene haploinsufficiency, thus confirming its deleterious effect. Conversely, for the KBG-like PT13 with a duplication partially involving the *ANKRD11* gene initially classified as likely pathogenic, the transcript analysis found the expected expression level, prompting us to re-classify the CNV as VUS. In a recent study where 23 patients were investigated for *ANRKD11* defects [23], two deletions in IVS1 were reported as VUS, while a partial gene duplication was classified as pathogenic. As proven in our study, the application of RT-qPCR analysis in these patients could impact the clinical diagnosis, particularly in individuals with a non-classic KBGS phenotype who may present with non-specific syndromic ID. Moreover, our preliminary data confirm the utility of this approach in agreement with a strong KBGS clinical diagnosis, as we found *ANKRD11* haploinsufficiency only in KBG patients (PT4, PT12, and PT14), and not in the KBG-like patient (PT13) where *ANKRD11* involvement was excluded.

In 18 KBG patients who were confirmed after NGS screening, we identified 17 different nucleotide variants, including seven nonsense, nine indels (one shared by monozygotic twin sisters), and one missense (Table 1). Seven variants were novel, and the others were previously reported in KBGS patients or in genomic databases such as dbSNP, ClinVar, and Decipher. All the 16 inactivating variants are considered pathogenic as their transcripts are predicted to encode proteins with different length and domain composition, or to lead to gene haploinsufficiency through the activation of nonsense-mediated decay. The pathogenicity of missense variants is challenging to envisage and is mainly based on bioinformatics prediction with a VUS/likely benign/benign classification, as is emerging from Varsome and gnomAD databases [30,31]. Although nonsynonymous variants in *ANKRD11* represent an infrequent defect in KBGS, a recent study reported 21, mostly de novo, *ANKRD11* missense variants whose predicted pathogenic effect was analyzed through functional approaches [25]. Most of the variants, clustered in Repression Domain 2 (RD2), seemed to cause reduced ANKRD11 stability and decreased proteasome degradation. Our p.Arg2536Trp missense variant (PT27) lies in this domain and is classified as likely pathogenic, according to ACMG guidelines, even if its effect needs to be further investigated.

With regard to the phenotypes associated with these molecular defects, the clinical manifestations are highly variable (Appendix A). They include classic KBGS features, subtle phenotypes, and non-specific signs shared by neurodevelopmental disorders. To evaluate a correlation between some clinical features and *ANKRD11* defect, we performed a comparison of the main clinical signs exhibited by the KBG-confirmed patients and the 11 negative cases. The latter group included five patients who resulted negative in all molecular tests, five patients with missense variants, and the patient with the genomic duplication as no alteration in *ANKRD11* expression that emerged from our gene expression study. The analysis did not detect any statistical differences between the two groups (Appendix A) and showed only a prevalence of three clinical signs in the KBG-confirmed patients, namely low hairline, synophrys, and hearing loss. The lack of significance highlights how the global phenotype associated with KBGS may be highly variable, subtle, mild, or difficult to be clearly identified. In addition, it is well-known that KBGS shares several clinical signs with other syndromes such as Cornelia De Lange, Coffin–Siris, Silver–Russell, and MRD23. Moreover, several NDD phenotypes that do not meet a KBG clinical diagnosis do present an alteration in *ANKRD11*, as proven by its high mutation rate in DDD studies identified through massive parallel sequencing [6,7,8,9]. These pieces of evidence emphasize the challenges in making a clinical diagnosis and imply an underestimation of KBGS. An additional contributor to the missed diagnosis is the presence of *ANKRD11* cryptic molecular defects that elude the standard molecular tests lowering the efficiency of the diagnostic approach currently used.

For all the above reasons, we suggest the use of large (physical or virtual) NGS panels, including NDD genes, in order to expedite the molecular diagnosis in these patient groups, to adopt exon-focused CMA analysis with increased resolution in the promoter region, and to perform gene expression analysis. Our data demonstrate that the assessment of *ANKRD11* haploinsufficiency by RT-qPCR is a helpful tool for molecular and clinical diagnosis both in patients with *ANKRD11* sub-microscopic rearrangements in whom the molecular effect is uncertain, and in molecularly unsolved KBGS patients in whom recognized haploinsufficiency might foster further studies targeted to the promoter region. We believe that the application of RT-qPCR analysis in the diagnostic flowchart for suspected KBGS patients will increase the diagnostic yield and expand the knowledge about the molecular defects underlying this syndrome.

## 4. Materials and Methods

### 4.1. Patients

Thirty-three patients, 18 males and 15 females, including two monozygotic twin sisters, with a clinical suspicion of KBGS were recruited as part of an international collaboration between investigators of Italy, United Kingdom, and Norway. Age at enrollment into the study ranged from 1 to 30 years (mean 12 ± 6 years). All patients underwent complete physical examination by their referring clinical geneticists. Intellectual disability and developmental delay of varying degrees, learning disabilities, and/or behavioral problems were present in the majority of patients together with at least two major criteria or one major and two minor criteria, according to the diagnostic aid reported by Low et al. [5]. The clinical manifestations of the reported patients are summarized in Appendix A. Informed consent was obtained from all individuals and/or their legal guardians enrolled in this study. An additional consent was obtained for the publication of photographic material that identify the patients.

In patients with a clinical suspicion of KBGS, the first-tier test was *ANKRD11* sequencing through an NGS-based multi-gene panel, including genes for NDDs that should be considered in the differential diagnosis. For individuals in whom no pathogenic variants were identified, CMA, preferably at a high resolution, was performed. A specific *ANKRD11* gene expression analysis was carried out when no rare Copy Number Variants (CNVs) were detected or where CNVs involving *ANKRD11* were of uncertain significance in order to look for cryptic genetic defects or CNVs undetected by the previous tests (Figure 1).

### 4.2. ANKRD11 Sequencing

*ANKRD11* sequencing (NM_013275.6) was performed on all but two patients. Sequencing of blood DNA made use of different techniques, according to the standards of each laboratory. In three patients (PT1, PT2, and PT3), all the exons and intron-exon junctions of the *ANKRD11* gene were screened by Sanger sequencing, following standard protocols [24]. Primer sequences and PCR conditions are available upon request. Twenty-seven patients underwent targeted next-generation sequencing (NGS) with the use of different gene panels, including *ANKRD11*: 26 patients were tested with a diagnostic targeted NGS panel, including the KBGS and CdLS-related genes, based on a custom Nextera Rapid Capture enrichment kit (Illumina, San Diego, CA, USA), and PT14 was tested with the TruSight One Sequencing Panel (Illumina), including 4813 genes. In PT23, the twin sister of PT22, a targeted PCR followed by Sanger sequencing was performed, while exome sequencing was performed on PT31 using the Agilent SureSelect Clinical Research Exome enrichment kit (Agilent, Santa Clara, CA, USA), as previously described [32].

Briefly, variants with a minor allele frequency (MAF) >1% (based on 1000 Genome project and gnomAD frequency) and synonymous variants were excluded from the analysis. All remaining variants were investigated through bioinformatics prediction tools and checked in public and licensed databases (HGMD professional v.2021.4, LOVD v.2.0, and Varsome, accessed February 2022) [30,33]. Variant classification was accomplished according to the guidelines of the American College of Medical Genetics and Genomics and the Association for Molecular Pathology (ACMG/AMP) [34]. Validation and segregation of selected variants detected by NGS approaches were performed on the patients and their parents by Sanger sequencing.

### 4.3. High-Resolution Chromosome Microarray Analysis

High-resolution CMA was performed on 10 patients and their parents, when available, using the Human Genome CGH Microarray Kits 2 × 400 K (Agilent), according to the manufacturer’s instructions. In two patients (PT4 and PT14), a further analysis was performed using the Human Genome CGH Microarray Kits 1 × 1 M (Agilent). GentiSure Postnatal Microarray 4 × 180 K+SNP ISCA (Agilent) was used in PT33. Data extraction and analysis were performed using CytoGenomics v.3.0 (Agilent). In PT12 and his parents, CMA was performed using the OGT CytoSure Constitutional v3 8 × 60 k array (Oxford Gene Technology, Oxfordshire, UK), which is designed for genes involved in developmental delay/learning difficulties, with probe enrichment at syndromic regions. Data were analyzed using the CytoSure Interpret version 4.8.32 (Oxford Gene Technology). The guidelines suggested by Miller et al. [26] and subsequently by the ACMG [27] were followed for CNV classification, with minor modifications.

### 4.4. Evaluation of ANKRD11 RELATIVE Expression

The total RNA of 10 patients, their parents, and 10 healthy controls was collected using Tempus Blood RNA tubes (Thermo Fisher Scientific, Waltham, MA, USA), isolated using the Tempus Spin RNA Isolation kit (Thermo Fisher Scientific) and reverse-transcribed using the High-Capacity cDNA Reverse Transcription kit (Thermo Fisher Scientific). Quantitative real-time RT-PCR (RT-qPCR), based on the TaqMan methodology, was performed using an ABI PRISM 7700 Sequence Detection System (Applied Biosystems, Foster City, CA, USA). Three *ANKRD11* (NM_001256182.1) TaqMan assays mapping exon junctions 2–3 (ID#Hs00203193_m1), 8–9 (ID#Hs00946581_g1), and 12–13 (ID#Hs00946585_gH) were used. The amounts of *ANKRD11* mRNA were calculated using the 2^−ΔΔCt^ method [35], with *GUSB* (ID#Hs00939627_m1) and *TBP* (ID#Hs99999910_m1) as the endogenous normalizing genes. All assays were provided by Thermo Fisher Scientific. Real-time data was analyzed using the RQ Manager 1.2 software (Applied Biosystems). We established the proper range of gene expression in 10 healthy controls by calculating the mean value ±2 standard deviation (SDS). If the expression level in the patient was below the control range, *ANKRD11* downregulation was inferred.

### 4.5. CNVs Molecular Characterization

For PT4 and PT14’s DNA, quantitative PCR assays (qPCR) were carried out using the SYBR green methodology with specific *ANKRD11* primer pairs, as previously described [36]. CNV junction characterization through Long-Range PCR was performed on PT4’s DNA using the TaKaRa LA Taq™ kit (Takara Bio Inc., Shiga, Japan), while the KAPA2G™ Robust PCR Kit (Kapa Biosystems, Wilmington, MA, USA) was used for PT14’s DNA. Amplicons were sequenced by direct dideoxy sequencing and then analyzed, as previously described [36]. Primers used for CNV characterization are listed in Appendix A.

### 4.6. Statistical Analysis

To evaluate significant differences between the main clinical signs reported in molecularly confirmed and unconfirmed KBGS patients, Fisher’s exact test was performed using R.3.1.0 software. The 2 × 2 contingency tables were used, and a value of *p* < 0.05 denoted a statistically significant test.

## Figures and Tables

**Figure 1 ijms-23-05912-f001:**
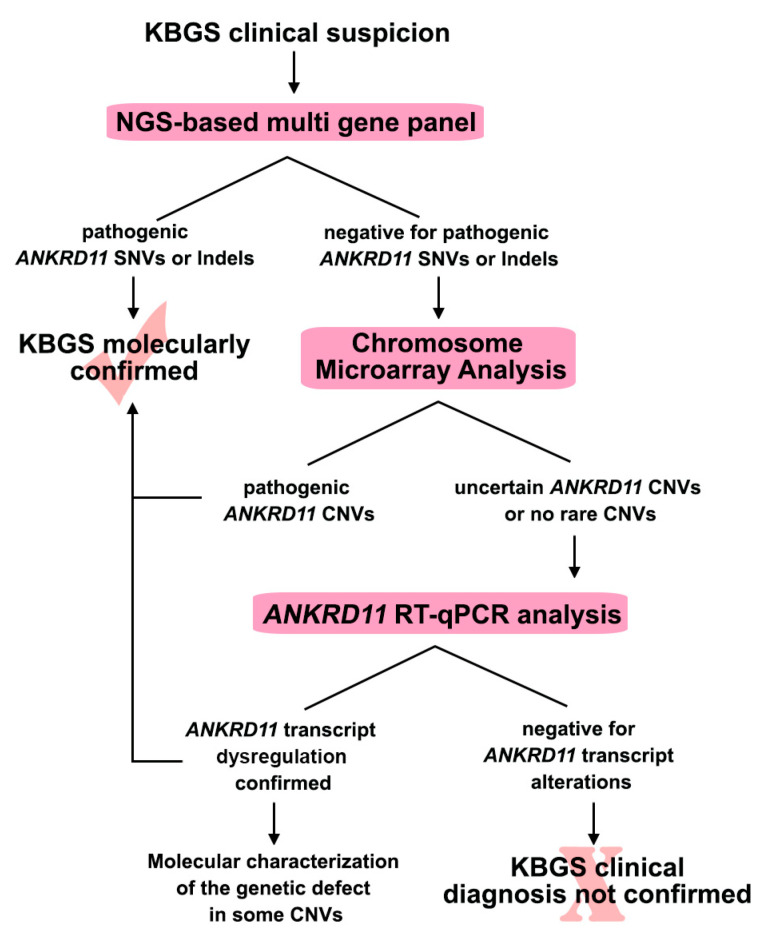
The KBGS flowchart. The figure summarizes the molecular diagnostic work-up that was used in the present study and which led to the molecular diagnosis in our KBGS patients. SNV: Single Nucleotide Variation; CNV: Copy Number Variations.

**Figure 2 ijms-23-05912-f002:**
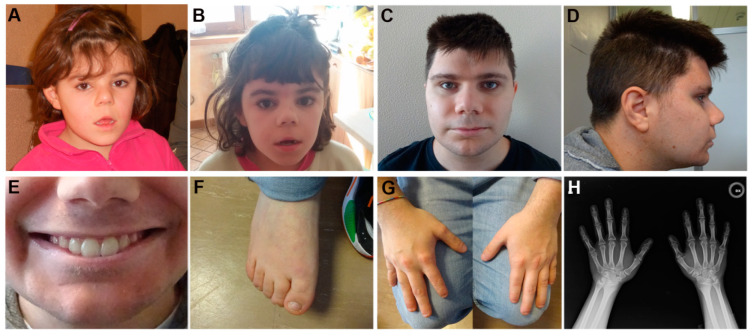
Clinical features of patients PT4 and PT33. (**A**,**B**) Facial features of PT4 at 3 and 9 years of age. (**C**,**D**) Frontal and profile facial features of PT33 at age 23 years. (**E**) Secondary dentition of PT33 showing wide upper central incisors (11 mm). (**F**) Foot of PT33 with large hallux. (**G**,**H**) Small hands of PT33 with hypoplasia of the 1st and 5th metacarpal bones and of the intermediate phalanx of the 2nd finger, bilaterally.

**Figure 3 ijms-23-05912-f003:**
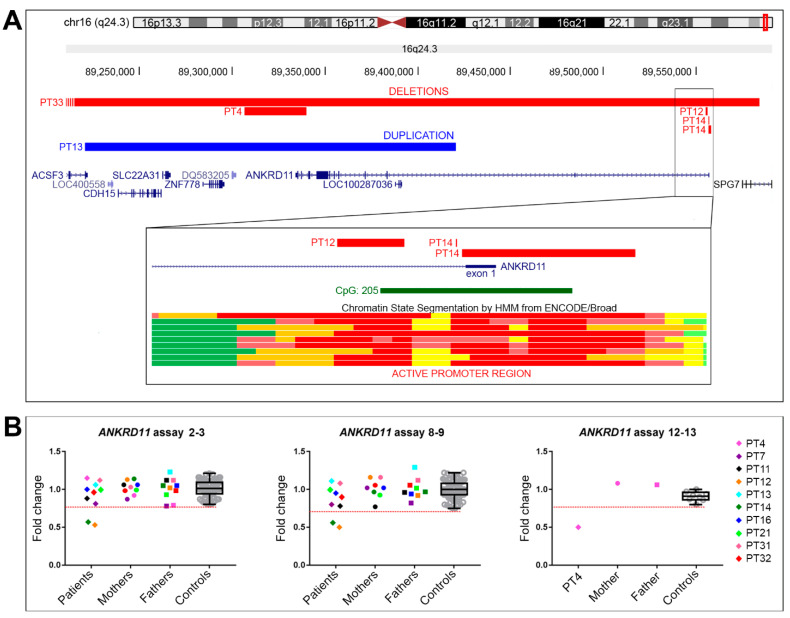
*ANKRD11* structural variants detected by CMA and gene expression analysis (RT-qPCR). (**A**) Physical map of the 16q24.3 region showing deletions of patients PT4 and PT33 as horizontal red bars, while the duplication of PT13 as a horizontal blue bar. Deletions of PT12 and PT14 are shown enlarged in the box. The genes mapping in this region are in blue characters, the CpG island in green characters, the predicted regulatory elements identified by ENCODE are represented by bars of different colors based on their function, with the active promoters depicted in red. The images are a modified version obtained from the UCSC Genome Browser (human genome assembly GRCh37/hg19). (**B**) RT-qPCR experiments performed on 10 patients (PT4, PT7, PT11, PT12, PT13, PT14, PT16, PT21, PT31, and PT32) with TaqMan probes for exon junctions 2–3 (all patients), 8–9 (all patients except PT4), and 12–13 (only PT4). Patients’ expression levels are shown as colored triangles, mothers and fathers as colored circles and squares, respectively. The transcript levels of healthy controls are shown as grey circles, and a boxplot sums up the *ANKRD11* expression variability. The dotted red line indicates the lower normal range cut-off.

**Figure 4 ijms-23-05912-f004:**
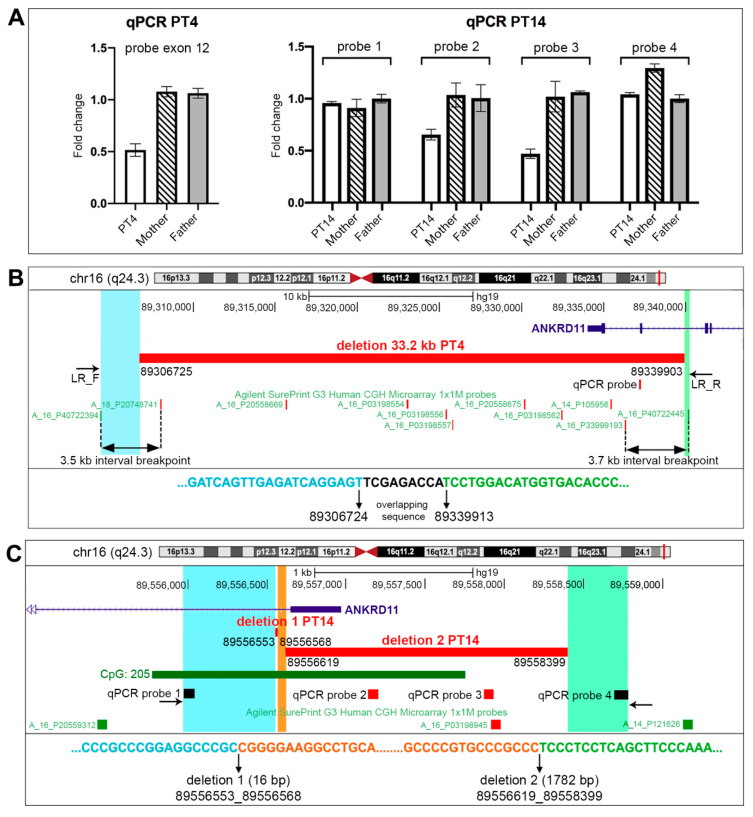
CNV characterization of PT4 and PT14. (**A**) Quantitative PCR on gDNA. qPCR results for PT4 and PT14, using probes spanning specific *ANKRD11* genomic regions to characterize the CNVs. The mother and father of each patient were used as normal controls. (**B**) Physical map of PT4’s deletion. The specific deletion of 33.2 kb with the relative nucleotide position is shown as a horizontal red bar. The qPCR probe on exon 12 and the Long-Range primers LR_F and LR_R used to redefine the CNV at the nucleotide level are shown. Interval breakpoints (black double-end arrows) are delimited by the CMA probes, which are shown in green if not deleted, and in red if deleted. The sequenced regions are highlighted in light blue and light green, with the breakpoint corresponding to the nucleotide sequence depicted below; a micro-homology region is indicated in black characters. (**C**) Physical map of PT14’s deletions. The specific deletions with the corresponding nucleotide breakpoint positions are shown as red bars. The CMA probes are shown in green if not deleted, and in red if deleted. qPCR probes are depicted in black or in red if deleted. The sequenced regions using qPCR probe 1 forward and qPCR probe 4 reverse are highlighted in light blue, orange, and light green. The corresponding nucleotide sequence is depicted below, and the breakpoints leading to the formation of the two 16 bp and 1782 bp adjacent deletions involving the promoter region of *ANKRD11* are indicated by the arrows.

**Table 1 ijms-23-05912-t001:** Variants identified through *ANKRD11* sequencing in KBG patients.

Patient ID	Gender	Sequencing Method	Chromosomal Position (hg19)	Exon	HGVS Nomenclature *	Variant Type	Inheritance	ACMG Classification ^#^	rs Number (dbSNP)	Allele Frequency (GnomAD)	Novelty	PMID
**Stopgain and Indel variants**												
PT1	M	Sanger Sequencing	chr16:89350427	9	c.2523G > A, (p.Trp841Ter)	Stopgain	De novo	P	unreported	unreported	This cohort	-
PT2	F	Sanger Sequencing	chr16:89346392	9	c.6552_6558dupTGAGGAG, (p.Pro2187Ter)	Insertion	De novo	P	unreported	unreported	This cohort	-
PT5	M	NGS gene panel	chr16:89349628	9	c.3319_3322delAAAG, (p.Lys1107AlafsTer210)	Deletion	Unknown	P	unreported	unreported	This cohort	-
PT6	F	NGS gene panel	chr16:89348689	9	c.4261G > T, (p.Glu1421Ter)	Stopgain	De novo	P	unreported	unreported	This cohort	-
PT9	F	NGS gene panel	chr16:89346113	9	c.6836_6837delTG, (p.Val2279GlyfsTer16)	Deletion	De novo	P	rs1555525296	unreported	Reported ^$^	-
PT15	F	NGS gene panel	chr16:89351563	9	c.1388_1389delAA, (p.Lys463ArgfsTer29)	Deletion	De novo	P	unreported	unreported	This cohort	-
PT17	M	NGS gene panel	chr16:89351043	9	c.1903_1907delAAACA, (p.Lys635GlnfsTer26)	Deletion	De novo	P	rs886041125	unreported	Reported	25413698
PT18	M	NGS gene panel	chr16:89350538	9	c.2408_2412delAAAAA, (p.Lys803ArgfsTer5)	Deletion	Unknown	P	rs886039902	unreported	Reported	27667800
PT19	F	NGS gene panel	chr16:89350753	9	c.2197C > T, (p.Arg733Ter)	Stopgain	Unknown	P	rs886041791	unreported	Reported	31191201
PT20	M	NGS gene panel	chr16:89351491	9	c.1459G > T, (p.Glu487Ter)	Stopgain	De novo	P	unreported	unreported	This cohort	-
PT22 ^‡^	F	NGS gene panel	chr16:89351566	9	c.1381_1384delGAAA, (p.Glu461GlnfsTer48)	Deletion	De novo	P	unreported	unreported	Reported	27605097
PT23 ^‡^	F	Sanger Sequencing	chr16:89351566	9	c.1381_1384delGAAA, (p.Glu461GlnfsTer48)	Deletion	De novo	P	unreported	unreported	Reported	27605097
PT24	M	NGS gene panel	chr16:89349181	9	c.3768_3769delCA, (p.His1256GlnfsTer26)	Deletion	De novo	P	unreported	unreported	Reported ^$^	-
PT26	F	NGS gene panel	chr16:89350549	9	c.2398_2401delGAAA, (p.Glu800AsnfsTer62)	Deletion	Unknown	P	rs797045027	unreported	Reported	25464108
PT28	M	NGS gene panel	chr16:89351578	9	c.1372C > T, (p.Arg458Ter)	Stopgain	Unknown	P	rs900492387	ƒ = 7.1 × 10^−6^	Reported	30202406
PT29	F	NGS gene panel	chr16:89350582	9	c.2368G > T, (p.Glu790Ter)	Stopgain	Unknown	P	unreported	unreported	This cohort	-
PT30	M	NGS gene panel	chr16:89351718	9	c.1232C > A, (p.Ser411Ter)	Stopgain	Unknown	P	unreported	unreported	Reported	32056211
**Missense variants**												
PT3	M	Sanger Sequencing	chr16:89349963	9	c.2987G > T, (p.Gly996Val)	Missense	Maternal	LB	rs1205687342	ƒ = 3.99 × 10^−6^	This cohort	-
PT8	F	NGS gene panel	chr16:89348536	9	c.4414G > A, (p.Glu1472Lys)	Missense	Paternal	LB	rs1597451653	unreported	This cohort	-
PT10	F	NGS gene panel	chr16:89347549	9	c.5401G > A, (p.Glu1801Lys)	Missense	Maternal	LB	rs938676909	unreported	This cohort	-
PT21	M	NGS gene panel	chr16:89345919 chr16:89346516	99	c.7031T > C, (p.Leu2344Pro) c.6434C > T, (p.Thr2145Ile)	Missense Missense	Maternal Maternal	US B	unreportedrs 761862402	unreported ƒ = 3.7 × 10^−5^	This cohort This cohort	-
PT25	M	NGS gene panel	chr16:89347752	9	c.5198C > T, (p.Ala1733Val)	Missense	Maternal	B	rs148243995	ƒ = 4.4 × 10^−4^	Reported ^$^	-
PT27	F	NGS gene panel	chr16: 89341329	11	c.7606C > T, (p.Arg2536Trp)	Missense	De novo	LP	unreported	unreported	Reported ^†^	-

Abbreviations are as follows: HGVS = Human Genome Variation Society, ACMG = American College of Medical Genetics and Genomics. * HGVS nomenclature applies to GenBank: NG_032003, GenBank: NM_013275.6, and GenBank: NP_037407. # classifications include pathogenic (P), likely pathogenic (LP), uncertain significance (US), likely benign (LB), and benign (B) variants. ‡ monozygotic twin sisters. $ variants reported in the ClinVar database. † variant reported in Decipher PT#304532 and in Boer et al., 2021 [25].

**Table 2 ijms-23-05912-t002:** Copy Number Variants identified through CMA analysis in the KBG cohort.

Patient ID	Gain/Loss	CNV Description According to the ISCN Nomenclature *	Size	Classification ^#^	Platform	Origin	Involved OMIM Genes
PT4	Loss	arr[GRCh37] 16q24.3(89307972-89335487) × 1 dn	27.5 kb	P	Agilent	De novo	*ANKRD11*
PT12	Loss	arr[GRCh37] 16q24.3(89555339-89556020) × 1 dn	682 bp	LP	OGT	De novo	*ANKRD11*
PT13	Gain	arr[GRCh37] 16q24.3(89220725-89420725) × 3	200 kb	LP	Agilent	Unknown	*ACSF3*, *CDH15*, *ANKRD11*
PT33	Loss	arr[GRCh37] 16q24.3(88365786-89584412) × 1	1.2 Mb	P	Agilent	Unknown	*ZNF469*, *CYBA*, *MVD*, *CTU2*, *PIEZO1*, *CDT1*, *APRT*, *GALNS*, *TRAPPC2L*, *ACSF3*, *CDH15*, *ANKRD11*

* International System for Human Cytogenomic Nomenclature (ISCN 2016). # The guidelines of Miller et al. [26] and of the ACMG [27] were followed for CNV classification.

## Data Availability

The data underlying this article are available in the article and in its online Appendix A. The list of SNVs/indels and CNVs identified in this cohort during the current study are publicly available at ClinVar database (http://www.ncbi.nlm.nih.gov/clinvar/): accession numbers SCV002097343 to SCV002097371.

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
