# Peer review of "Expanding the Molecular Spectrum of ANKRD11 Gene Defects in 33 Patients with a Clinical Presentation of KBG Syndrome"

_ijms, 2022, doi:10.3390/ijms23115912_

Round 1

Reviewer 1 Report

The manuscript submitted by Bestetti et al. expands the molecular catalog of gene defects associated with the KGB syndrome, reporting 12 new sequence variants. A significant contribution of this work is the incorporation of RT-qPCR to the molecular diagnosis flowchart to assess the pathogenic effect of genomic imbalances or to detect haploinsufficiency in molecularly unsolved patients. However, a few minor changes are needed before the manuscript is suitable for its publication in the International Journal of Molecular Sciences. The suggested changes are listed below:

  • Line 100: Substitute truncating by truncated
  • Line 101: The pathogenic variants cluster in exon 9 or 10? Correct if necessary or if it is correct, explain why almost all the variants detected in this study are located in exon 10.
  • Define the abbreviations SCV and CNV in the legend of Figure 1.
  • The title of Figure 3 must include targeted qPCR because this technique was used to characterize the molecular defect represented in panel A.

Author Response

Point by point responses to the Reviewers International Journal of Molecular Science - Molecular Genetics and Genomics section

(Pages and lines are referred to the clean MS)

Reviewer(s)' comments to Author

Reviewer #1:

Line 100: Substitute truncating by truncated

Amended

Line 101: The pathogenic variants cluster in exon 9 or 10? Correct if necessary or if it is correct, explain why almost all the variants detected in this study are located in exon 10.

Thanks the reviewer to point out the question. In the manuscript we referred to a different isoform (NM_001256182.1) and not the canonical one. To avoid misunderstanding we now refer to the canonical isoform (NM_013275.6) indicating in the text the mutation in exon 9 (see Table 1 revised, Table  S2  and S5,  Fig3, Fig 4  and Fig S1 revised, figure legends, and lines 127-128, line 132, line 136, lines 176-177, line 180, line 203-204, line 215, line 230, line 394, lines 438-439).

Define the abbreviations SCV and CNV in the legend of Figure 1.

The abbreviations SNV and CNV have been added at the end of the legend of Figure 1 (lines 123-124).

The title of Figure 3 must include targeted qPCR because this technique was used to characterize the molecular defect represented in panel A.

The title of Figure 3 has been revised as suggested (line 195).

Reviewer 2 Report

This manuscript reports the molecular studies performed on 33 individuals with clinical suspicion of KBG syndrome. By using sequencing, chromosome microarray analysis (CMA), and gene expression analysis, the Authors reached a molecular diagnosis in 22 out of 33 patients.

Lines 100-102  The Authors state: "Truncating pathogenic variants (frameshift or nonsense) are prevalent, and cluster mostly in exon 9 likely because it represents more than 80% of the coding region". This is in agreement with Ensembl (www.ensembl.org) which shows that the ANKRD11 gene has 13 exons including a very long (6578 nucleotides) exon 9. However, according to Table 1 most of the identified variants are located in exon 10, because the Authors use a different reference sequence where exon 9 is indicated as exon 10. This is also evident looking at Figure 3B, where the ANKRD11 assay named 12-13 (in the image), detects exon junctions 13-14 (in the legend).

The missense variant identified in PT27 is classified as pathogenic because one in silico tool predicts the activation of a cryptic donor splice site and the production of a truncated protein. In my opinion the Authors did not provide sufficient evidence to support this conclusion.

English language should be improved and some sentences should be rewritten more clearly. For example, lines 136-138: “the SNV might activate an exonic cryptic donor site, causing the skipping of 36 residues from exon 12 and the introduction 30 aa downstream of a premature stop codon in exon 13”; residues refer to the protein, not to the nucleotide sequence.

Lines 276-277: “The diagnostic yeld (yield) by NGS technology was 82% (18/22)” This is not correct, since for some patients the pathogenic variant was detected by Sanger sequencing.

Table 1 should be properly formatted.

Author Response

Point by point responses to the Reviewers International Journal of Molecular Science - Molecular Genetics and Genomics section

(Pages and lines are referred to the clean MS)

Reviewer(s)' comments to Author

Reviewer #2:

Lines 100-102 The Authors state: "Truncating pathogenic variants (frameshift or nonsense) are prevalent, and cluster mostly in exon 9 likely because it represents more than 80% of the coding region". This is in agreement with Ensembl (www.ensembl.org) which shows that the ANKRD11 gene has 13 exons including a very long (6578 nucleotides) exon 9. However, according to Table 1 most of the identified variants are located in exon 10, because the Authors use a different reference sequence where exon 9 is indicated as exon 10. This is also evident looking at Figure 3B, where the ANKRD11 assay named 12-13 (in the image), detects exon junctions 13-14 (in the legend).

Thanks the reviewer to raise this important point. As you noticed, we did not refer to the canonical isoform (NM_013275.6) but the isoform NM_001256182.1. To avoid misunderstanding we now refer to the canonical isoform and modified the text (lines 127-128, line 132, line 136, lines 176-177, line 180, line 203-204, line 215, line 230, line 394, lines 438-439), fgure legends and figures ( Figure 3, 4 and S1) and Table 1, Table  S2  and S5   revised accordingly.

The missense variant identified in PT27 is classified as pathogenic because one in silico tool predicts the activation of a cryptic donor splice site and the production of a truncated protein. In my opinion the Authors did not provide sufficient evidence to support this conclusion.

We agree with the reviewer. The p.Arg2536Trp missense variant is predicted to act as LoF mechanism, based on of the activation of an exonic donor splice site leading to the introduction of a premature stop codon (Figure S1). Interestingly the same variant has been functionally demonstrated to cause reduced ANKRD11 stability by Bear et al [25]. We clarified this issue in discussion ( lines  335-338)

English language should be improved and some sentences should be rewritten more clearly. For example, lines 136-138: “the SNV might activate an exonic cryptic donor site, causing the skipping of 36 residues from exon 12 and the introduction 30 aa downstream of a premature stop codon in exon 13”; residues refer to the protein, not to the nucleotide sequence.

The manuscript has been revised by an active English author. According to the useful reviewer suggestions we have reworded some sentences including that signaled by the reviewer. Lines 138-141: Based on the prediction, the SNV might activate an exonic cryptic donor site, causing the skipping of 36 residues encoded by  exon 11 and the introduction 30 aa downstream of a premature stop codon in exon 12 (Figure S1).

Lines 276-277: “The diagnostic yeld (yield) by NGS technology was 82% (18/22)” This is not correct, since for some patients the pathogenic variant was detected by Sanger sequencing.

We thank the reviewer and accordingly changed “NGS technology” to  “ANKRD11 sequencing” thus including both NGS and Sanger approaches (lines 279-280).

Table 1 should be properly formatted.

We formatted Table 1 according to Journal guidelines. As suggested, we have now  re-formatted Table 1 decreasing the font.

Reviewer 3 Report

The paper addresses a rare disorder and brings valuable data with regard to genetic etiology and specificity of clinical symptoms associated with ANKRD11 defects. In addition, the proposed investigation algorithm proved highly efficient for accurate classification of CNVs leading to a better understanding of genetic architecture of KBGS.

The paragraph between lines 87-94 would benefit from an additional reference, as ref 15 doesn't include Silver Russel syndrome. Moreover, SRS is not simply caused by a chromatin modifier gene, but instead there is an aberant chromatin architecture of the imprinted gene clusters at play.

Uniform ANKD11 exons numbering - it seems that exon 9 from introduction section is the same as exon 10 in the results section, with different NM used as reference. 

There is a minor editing error: line 80 - fontanelle instead of fontanel

Author Response

Point by point responses to the Reviewers International Journal of Molecular Science - Molecular Genetics and Genomics section

(Pages and lines are referred to the clean MS)

Reviewer(s)' comments to Author

Reviewer #3:

The paragraph between lines 87-94 would benefit from an additional reference, as ref 15 doesn't include Silver Russel syndrome. Moreover, SRS is not simply caused by a chromatin modifier gene, but instead there is an aberant chromatin architecture of the imprinted gene clusters at play.

We thank the reviewer for this comment and accordingly we mention separately SRS adding the following sentence: In addition, a patient carrying ANKRD11 deletion, with a normal psychomotor development and clinical signs reminiscent of Silver-Russell syndrome (SRS; OMIM#180860), has been reported [10] ( lines 86-96).

Uniform ANKD11 exons numbering - it seems that exon 9 from introduction section is the same as exon 10 in the results section, with different NM used as reference.

Thanks the reviewer to point out the question. As you noticed, we did not refer to the canonical isoform (NM_013275.6) but to isoform NM_001256182.1. To avoid misunderstanding we now refer to the canonical isoform and modified the text (lines 127-128, line 132, line 136, lines 176-177, line 180, line 203-204, line 215, line 230, line 394, lines 438-439), fgure legends and figures ( Figure 3, 4 and S1) and Table 1, Table  S2  and S5  revised accordingly.

There is a minor editing error: line 80 - fontanelle instead of fontanel" has been amended (line 80).

Round 2

Reviewer 2 Report

Unfortunately, also this revised version of the manuscript did not provide evidence that the missense variant detected in PT27 is pathogenic. The Authors wrote (lines 335-338) "Our p.Arg2536Trp missense variant (PT27) in this domain, predicted to activate an exonic donor splice site with the introduction of a premature stop codon (Figure S1), has been functionally demonstrated to cause reduced ANKRD11 stability [25]."  Ref.25 is a report on medRxiv, which is not a peer-reviewed journal. The Authors should either provide convincing evidence to support their conclusions, or classify the variant as VUS.

Author Response

We agree with the referee that further functional studies are needed to established the pathogenic effect of the p.Arg2536Trp missense variant. According to the ACMG guidelines, used to clinically classify all the variants identified in this study, the variant has been classified as likely pathogenic (see Table S2).

Therefore we modified the sentences in discussion lines x-y:  Most of the variants, clustered in Repression Domain 2 (RD2), seem to cause reduced ANKRD11 stability and decreased proteasome degradation. Our p.Arg2536Trp missense variant (PT27) lies in this domain and according to ACMG guidelines is classified as likely pathogenic even if its effect needs to be further investigated.

Round 3

Reviewer 2 Report

-